# Building Resilience from the Grassroots: The Cyclone Preparedness Programme at 50

**DOI:** 10.3390/ijerph192114503

**Published:** 2022-11-04

**Authors:** Ahmadul Haque, Dilruba Haider, Muhammad Saidur Rahman, Laila Kabir, Raul Perez Lejano

**Affiliations:** 1Cyclone Preparedness Programme, Dhaka 1212, Bangladesh; 2UN Women, Bangladesh Country Office, Dhaka 1212, Bangladesh; 3Bangladesh Disaster Preparedness Centre, Dhaka 1212, Bangladesh; 4School of Culture, Education, and Human Development, New York University, New York, NY 10012, USA

**Keywords:** disaster risk reduction, Cyclone Preparedness Programme, gender equality

## Abstract

As Bangladesh’s Cyclone Preparedness Programme (CPP) celebrates its 50th anniversary, we reflect on its legacy, the gains made, and progress that still lies ahead. The CPP is unique among disaster risk-management agencies, as more than 90% of its staff consists of community volunteers. This unique institutional design influences its functioning. An important part of its growth has included the expansion of the involvement and leadership of women in the CPP, so that today, women constitute about 50% of the volunteer corps. We reflect on the improvements made, since Cyclone Bhola of 1970 (the deadliest natural tragedy on record) and analyze institutional features of the CPP that other countries can learn from. Lastly, we reflect on remaining challenges for the years ahead.

## 1. Background

Bangladesh is one of the most disaster-prone countries in the world. It is affected by different types of natural hazards almost every year, mainly cyclones and floods, due to its geographical location and topography. The disasters cause the loss of human lives and physical assets, the effects of which are further reflected in social settings, ecosystems and the economic well-being of people. Poor socio-economic conditions and an extremely high population density add to the people’s high vulnerability to disasters.

Due to the unique configuration and bathymetry of its southern coast, Bangladesh has seen the most devastating storm surges in recent history. This vulnerability is compounded by challenging socio-economic conditions (in addition to the above, Bangladesh experiences other natural hazards such as drought, tornadoes, and riverbank erosion (Karim, 1995 [1])). Bangladesh’s mean per-capita GDP lies at USD 1829, which ranks among the lowest in the world’s nations; 23.2% of its population lies below the poverty line; and 12.0% lies below the extreme-poverty threshold. These vulnerabilities are magnified in areas prone to natural disasters, where literacy rates are lower and 33% of the population has received no education (BBS, 2019 [2]). Over the last five years, the southeast district of Cox’s Bazar has seen the entry of almost a million, highly vulnerable Rohingya refugees fleeing crossing the border with Myanmar, and these refugees constitute another highly vulnerable segment of the population.

The Bhola cyclone of 1970 was a devastating tropical cyclone that struck Bangladesh and India’s West Bengal on 12 November 1970. It remains the deadliest tropical cyclone ever recorded in the history of the world. Over 1,000,000 people lost their lives, primarily as a result of the storm surge that flooded much of the low-lying islands of the Ganges Delta. The cyclone of 1991 struck the Chittagong district with wind speed of around 225 km/h on the night of 29 April. The storm forced a 6-metre storm surge inland over a wide area, killing over 300,000 people.

## 2. The Cyclone Preparedness Programme

After the devastating Bhola cyclone of 1970, the Government of Bangladesh and the Bangladesh Red Crescent Society established the Cyclone Preparedness Programme in 1973. The large scope and population in its jurisdiction necessitated a unique organizational design. Its operations are led and run largely by a corps of 76,000 community volunteers spread out over 3801 village units. Its core essence is to nurture change within the vulnerable community by and for the community itself. Each village unit consists of 20 community volunteers, numbering 10 female and 10 male residents. The core functions of the volunteers consist of early-warning dissemination, combing the community in advance of a tropical cyclone, using megaphones, loudspeakers, and hand-held sirens. Following government advisories to evacuate, the volunteers then assist vulnerable residents (including persons with disabilities, frail elderly, expectant mothers, and children) to evacuate to cyclone shelters (CPP, 2021a [3]).

A primary goal of the paper is to highlight the uniqueness of the CPP and lessons that can be learned from the CPP by disaster risk-management agencies worldwide. Among other things, we discuss unique features of the CPP such as: the fact that the vast majority of its personnel consist of community volunteers, and its formal empowerment of women (to the extent of implementing a goal of 50% membership of women in the CPP). Another distinctive feature of the CPP is its implementation of a people-centered mode of risk communication wherein community residents are empowered to be risk communicators in their communities.

The CPP should be globally recognized as a distinctive model of community engagement. CPP volunteers maintain social recognition and standing in their communities. A distinctive achievement has been the fuller inclusion of women in the CPP volunteer corps, rising from a ratio of 1:6 (women to men) in 1992 to roughly equal numbers today. Parallel to this is the initiative to empower women to take on leadership roles in the organization (CPP, 2021a [3]).

The institution of the CPP has been identified as a key development in the building of resilience to disasters in Bangladesh. This has proceeded along with developmental progress, as seen in increasing per-capita GDP, literacy rates, and health indices. Thousands of cyclone shelters have been built. Mortality from cyclone hazards has decreased dramatically, to below a hundred over the last decade. Morbidity has dropped dramatically as well.

It would be good to, however cursorily, compare the CPP to similar agencies in other cyclone-prone countries in this region. In the case of India, disaster management falls under the purview of a national agency, the National Disaster Management Authority (NDMA), which extends to the state level (State Disaster Management Authority, or SDMA) and districts. This agency disseminates early warning at the lower levels and carries out evacuation in coordination with Civil Defense (INDMA, 2008 [4]). Similarly, the Philippines implements disaster management in a hierarchical fashion, from the National Disaster Risk Reduction and Management Council (NDRRMC) to Provincial Disaster Risk Reduction and Management Council (PDRRMC) to Local Disaster Risk Reduction and Management Council (LDRRMC), and shares responsibilities for disseminating early warnings with the Office of Civil Defense, while evacuation is mainly carried out by the LDRRMC (in coordination with local government units and community-based organizations). Similarly, Pakistan’s disaster-management program has a hierarchical structure, from the national NDMA to PDMA to DDMA, and early warnings are delivered through this chain of command. These agencies coordinate with Civil Defense, which works with local district governments in evacuation (PNDMA, 2010 [5]).

The CPP is distinctive in several respects. First, it is a single-purpose agency with a special focus on tropical cyclones. This may lend to a simpler organizational arrangement, in which all functions related to cyclone preparedness and response are centralized within the CPP. While we do not have enough data to analyze the organizational strengths and limitations of this more streamlined organizational design vis-a-vis other national agencies, suffice it to say that this arrangement tends to allow easier replicability from district to district.

Another distinctive feature of the CPP is its formal commitment to community-based disaster management (CBDM). Note that other nation’s agencies also have a commitment to CBDM, but in a less formal fashion. For example, in India, community volunteers are organized on an ad-hoc level—in some cases, through Civil Defense and, in others, through different local and regional civic organizations (INDMA, 2008, [4]). A similar arrangement is found in the Philippines, where various NGOs and CBOs coordinate with the Office of Civil Defense and LDRRMCs in a manner that differs from locality to locality. The CPP is unique in the extent to which community-based disaster management is formalized—most evident of all in the fact that most of the workforce of CPP is composed of formally deputized CPP volunteers. The formal deployment of community residents translates into a formal commitment to gender equity as well—the CPP is also unique among similar national agencies in its formal goal of 50% participation by women in the CPP volunteer corps (INDMA, 2008, [4]), as discussed below. It is quite possible that the strength of these commitments to community participation and gender equity are fostered by the clarity of CPP’s mandate and organization. Much literature is already available positively linking women’s participation in disaster risk management to improved outcomes such as improved outreach, greater use of shelters, and reduced fatalities (e.g., UNISDR, 2009 [6]). There is ample literature documenting the greater vulnerability of lower income women in parts of Bangladesh (e.g., Ikeda, 1995 [7]), and greater involvement and leadership of women in the CPP enables the organization to reach these vulnerable women more effectively (CPP, 2021b [8]). In addition, so, the CPP helps communities by empowering women within them—it is a two-fold solution. It has been repeatedly seen in disasters that women and girls are most affected in disasters due to socio-economically and culturally disadvantaged positions in society (e.g., Thurston et al., 2021 [9]). Therefore, to alleviate their plight, women’s engagement and leadership is imperative for a gender-responsive approach to disaster response.

The CPP, more so than other international agencies, has succeeded in creating a largely volunteer-led corps for early warning and emergency preparedness and in institutionalizing parity of participation by women. These also constitute ongoing challenges in maintaining the CPP. For example, there is often a discussion about whether the volunteer corps should be given, if not formal salaries, stipends, but this brings up the issue of whether participation would then be driven more by financial incentives than the motivation to help.

## 3. Progress Made

The most notable achievement of the CPP has to be the great reduction in the human toll of tropical cyclones. While this is due to many factors (including improved infrastructure and the construction of cyclone shelters), the increase in numbers of the CPP volunteers has been cited as a major reason for this. Figure 1 plots the declining fatalities associated with tropical cyclones over time, along with the steady rise in numbers of volunteers (and, again, improvements in disaster risk reduction are due to many factors and efforts, including those by other organizations).

Another notable step in the CPP’s evolution has been the recent expansion of its jurisdiction to include the sprawling Rohingya refugee camps in Cox’s Bazar. Since 2017, Rohingya refugees have flooded into Bangladesh at an alarming rate following persecution in their home country of Myanmar. The CPP has provided DRR activities inside the camp settlements since February 2018 with the support of BDRCS, IFRC and the American Red Cross. In this work, the CPP has made use of its orientation toward “multi-hazard switching,” which entails a flexibility to engage with differing risk scenarios. Unlike the work with coastal communities threatened by storm surges, the CPP’s work in the settlements dealt with risks due to mudslides, flash floods, and slope failure.

It is indeed important to understand that “disasters don’t discriminate, but people do disasters reinforce, perpetuate and increase gender inequality, making bad situations worse for women” (UNISDR, 2009, [6]). One historical review found that, in less developed countries, females had a higher risk of mortality during tropical cyclones (Doocy et al., 2013 [11]). One study (Nasreen, 2014 [12]) concludes that women in disaster-prone areas often suffer more from sexual- and reproductive-health-related problems during and after disasters. A study on violence against women in disasters (Nasreen, 2008 [13]) indicates that a large number (71.6%) of women (studied) were subject to more violence during disasters. Sexual harassment, including forced sex, and rape at home and in shelters, was also reported by women and girls; and they do not want to take refuge in shelters during disasters due to lack of privacy and security. At the same time, women’s actual and potential contributions to disaster risk reduction is seldom recognized. Women’s role in strengthening a sustainability culture and their active contribution to household and community stability during disasters warrant due recognition.

The Bangladesh Government recognizes the importance of gender-responsive disaster reduction actions. The key national instruments such as SOD (2019), and the latest National Plan for Disaster Management (NPDM) have well integrated aspects of gender equality and women’s empowerment. CPP have made ‘gender parity’ a key principle and have 50% female volunteers in their 74,000 volunteer group across the coastal belt of the country (CPP, 2021a [3]).

The co-authors have also collaborated on a Relational Model of Risk Communication wherein community residents are empowered to act as risk communicators themselves and bearers of risk knowledge. Training modules involve learning skills in translating weather bulletins into everyday language/narrative and strategizing how to spread risk knowledge in the community.

## 4. Policy Lessons and Challenges Ahead

The CPP is a singular case study that holds important institutional lessons for disaster risk-management agencies around the world. It has met the challenge of resource limitations by tapping into the great store of social capital in communities. To do this, it has instituted an ambitious, grassroots-based volunteer training program, deputizing tens of thousands of volunteers and constructing the volunteer corps as a mark of distinction for local residents. The full engagement of communities in disaster risk reduction has entailed a novel organizational structure, including teams of volunteers instituted down to the smallest geographic units. The CPP’s model of “people-centered action” is a policy model that other countries can learn from. 

The massive participation of communities poses ever-present policy and organizational challenges. Institutional processes (such as widespread and frequent training) are needed to maintain the large unsalaried volunteer corps, as well as maintaining the skills and knowledge base. More progress is needed in strengthening the volunteer corps, maintaining morale, and possibly instituting some modest amount of monetary compensation. The current generation of community volunteers were not part of the heady days of nation building that began in 1970, and there is a need to strengthen the sense of mission among them.

Another needed area of policy action is seeking the fuller inclusion of women in leadership roles. While women now constitute around 50% of the volunteer corps, there needs to be gains in promoting women to leadership positions in the CPP. The key recommendations are: (1) women leadership has to be enhanced; in addition to increasing the numbers of women in leadership positions; (2) women’s agencies need to be brought on board in the disaster reduction discussions; (3) sex–age and disability disaggregated data (SADDD) should be collected and disseminated; and (4) women’s security and protection, and their reproductive-health care, needs to be an integral part of humanitarian actions.

Another future area of improvement involves integrating technology into grassroots initiatives. There has been discussion of more strategic use of social media and cellphone texts for early warning. While there exists a nationwide GIS-based geohazard map, its use has not been integrated into the planning and operation of the CPP.

Overall, expansion of needed infrastructure such as additional typhoon shelters is needed in remote areas. Collaboration among the authors created what is known in the literature as a Relational Model of Risk Communication (Lejano et al., 2020 [14]; Lejano et al., 2022 [15]). Further progress on training residents to engage in this model of risk communication is needed, as such training has not been instituted in all coastal districts. Empowerment programs to ensure that new community volunteers have the same motivation as previous groups. However, all in all, Bangladesh’s CPP, in its 50 years, has proven to be a sustainable and innovative strategy for disaster preparedness.

## 5. Conclusions

Now in its fiftieth year, The Bangladesh Cyclone Preparedness Programme is a unique type of disaster risk management agency that underscores institutional efforts at community engagement and gender equality. The authors believe that its unique institutional features and effectiveness in the field offer much that agencies in other countries can learn from. At the same time, there is still much progress needed in the years ahead. 

## Figures and Tables

**Figure 1 ijerph-19-14503-f001:**
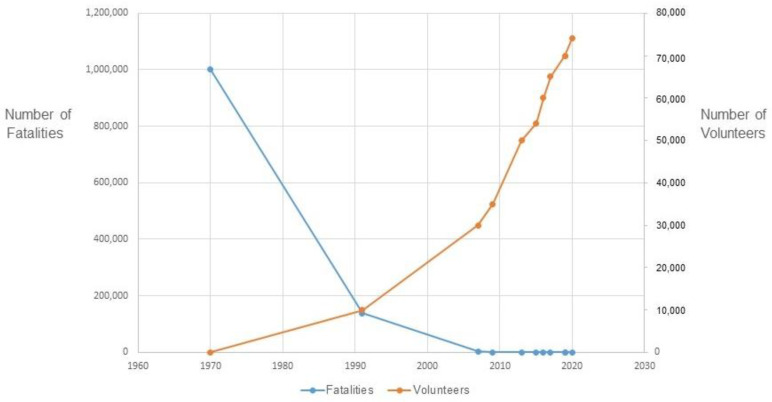
Historical trends in fatalities and volunteers due to tropical cyclones (Note: data for the figure was obtained from CPP, 2021c [10]).

## Data Availability

No data to report.

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
