# Peer review of "Building Resilience from the Grassroots: The Cyclone Preparedness Programme at 50"

_ijerph, 2022, doi:10.3390/ijerph192114503_

Round 1

Reviewer 1 Report

The article lacks a clearly written goal. There are no conclusions supported by a thorough analysis.

 Page 1, first paragraph: Are there any other natural hazards in Bangladesh besides cyclones and floods?

Page 2: Is this Program about helping people or is it about women taking leadership? Please write it more clearly and justify it. Will leadership by women solve the problem or not make it worse?

Fig 1: Explain Y axes, what does left mean and what means right?

The article lacks a thorough analysis of the phenomena and processes taking place. For example, isn't the fact that more women experience the effects of a cyclone because of the greater proportion of women in the population? Without such analyzes, the article is only a speculation by the authors.

 Is the preference for women to managerial positions not a symptom of discrimination against men? Should gender decide about performing managerial functions? Doubtful! Shouldn't you look for some other solutions here?

The article needs to be rethought. To discuss the problem a bit more.

Author Response

Response to Reviewer 1's Comments

The authors are grateful for the reviewer's comments and suggestions. As described below, we have endeavored to address all of them (while staying close to the journal's word limits for short articles).

Comment:  The article lacks a clearly written goal. There are no conclusions supported by a thorough analysis.

Response:  We have added the following text to the revised ms. The rest of the revised ms is written to support this goal.

A primary goal of the paper is to highlight the uniqueness of the CPP and lessons that can be learned from the CPP by disaster risk management agencies worldwide. Among other things, we discuss unique features of the CPP like: the fact that the vast majority of its personnel consist of community volunteers, and its formal empowerment of women (to the extent of implementing a goal of 50% membership of women in the CPP). Another distinctive feature of the CPP is its implementation of a people-centered mode of risk communication wherein community residents are empowered to be risk communicators in their communities.

Comment:  Page 1, first paragraph: Are there any other natural hazards in Bangladesh besides cyclones and floods?

Response:   We have added the following sentence and citation.

In addition to the above, Bangladesh experiences other natural hazards such as drought, tornadoes, and riverbank erosion (Karim, 1995).

Karim, N. (1995). Disasters in Bangladesh. Natural Hazards, 11(3), 247-258.

Comment:  Page 2: Is this Program about helping people or is it about women taking leadership? Please write it more clearly and justify it. Will leadership by women solve the problem or not make it worse?

Response:  We have added the following text and citations.

Much literature is already available positively linking women's participation in disaster risk management to improved outcomes such as improved outreach, greater use of shelters, and reduced fatalities (e.g., UNISDR, 2009). There is ample literature documenting the greater vulnerability of lower-income women in parts of Bangladesh (e.g., Ikeda, 1995), and greater involvement and leadership of women in the CPP enables the organization to reach these vulnerable women more effectively (CPP, 2021b). And, so, the CPP helps communities by empowering women within them --it is a two-fold solution. It has been repeatedly seen in disasters that women and girls are most affected in disasters due to socio-economically and culturally disadvantaged positions in society (e.g., Thurston et al., 2021). So, to alleviate their plight, women’s engagement and leadership is imperative for a gender responsive approach to disaster response. 

CPP, Bangladesh Cyclone Preparedness Programme (2021b). WE-CPP: Women Empowerment in Cyclone Preparedness Programme, Ministry of Disaster Management and Relief, Dhaka, Bangladesh.

Ikeda, K. (1995). Gender differences in human loss and vulnerability in natural disasters: a case study from Bangladesh. Indian Journal of Gender Studies, 2(2), 171-193.

Thurston, A. M., Stöckl, H., & Ranganathan, M. (2021). Natural hazards, disasters and violence against women and girls: a global mixed-methods systematic review. BMJ global health, 6(4), e004377.

UNISDR (2009). Making Disaster Risk Reduction gender sensitive: policy and practical guidelines, United Nations International Strategy for Disaster Reduction, Geneva, Switzerland.

Comment:  Fig 1: Explain Y axes, what does left mean and what means right?

Response:  We have improved Figure 1 by adding labels to the left and right Y axes.

Comment:  The article lacks a thorough analysis of the phenomena and processes taking place. For example, isn't the fact that more women experience the effects of a cyclone because of the greater proportion of women in the population? Without such analyzes, the article is only a speculation by the authors.Is the preference for women to managerial positions not a symptom of discrimination against men? Should gender decide about performing managerial functions? Doubtful! Shouldn't you look for some other solutions here?

Response:  As described above (and in much of the text in the revised ms), in fact, it has been well established in the DRR literature that increased particiption by women is associated with improved disaster risk reduction. In the case of the CPP, increasing the numbers of women involved is not discriminating against men because traditionally, there were much greater numbers of men involved in the CPP volunteer corps than women. The present policy simply aims at reducing the disparity and, so, it is not a discrimination against men but a reduction in the shortage of women in the program (since roughly half of the community members are women, there is a shortage when much less than half of the volunteers are women --this is the gap that is being addressed). The numbers of female victims compared to men have been statistically shown in numerous studies (too numerous to cite in a short paper with a 2500 word limit), and these statistics are relative to the proportion of females to males in the surrounding population (e.g. use of odds-ratios). This is not speculation but a statistically demonstrated phenomenon --see, for example, the modified text in the revised ms below.

It is indeed important to understand that “disasters don’t discriminate, but people do… disasters reinforce, perpetuate and increase gender inequality, making bad situations worse for women” (UNISDR, 2009, 17). One historical review found that, in less developed countries, females had a higher risk of mortality during tropical cyclones (Doocy et al., 2013). One study (Nasreen, 2014) concluded that women in disaster prone areas often suffer more from sexual and reproductive health related problems during and after disaster. A study on violence against women in disasters (Nasreen, 2008) indicates that a large number (71.6%) of women (studied) were subject to more violence during disasters. Sexual harassment including forced sex, rape at home and in shelters were also reported by women and girls; and they do not want to take refuge in shelters during disasters due to lack of privacy and security. At the same time, women’s actual and potential contributions to disaster risk reduction is seldom recognised. Women’s role in strengthening a sustainability culture and their active contribution to household and community stability during disasters warrant due recognition.

Lastly, the primary points of the paper are summarized for clarity in the following added text.

A primary goal of the paper is to highlight the uniqueness of the CPP and lessons that can be learned from the CPP by disaster risk management agencies worldwide. Among other things, we discuss unique features of the CPP like: the fact that the vast majority of its personnel consist of community volunteers, and its formal empowerment of women (to the extent of implementing a goal of 50% membership of women in the CPP). Another distinctive feature of the CPP is its implementation of a people-centered mode of risk communication wherein community residents are empowered to be risk communicators in their communities.

Response to Reviewer 1's Comments

The authors are grateful for the reviewer's comments and suggestions. As described below, we have endeavored to address all of them (while staying close to the journal's word limits for short articles).

Comment:  The article lacks a clearly written goal. There are no conclusions supported by a thorough analysis.

Response:  We have added the following text to the revised ms. The rest of the revised ms is written to support this goal.

A primary goal of the paper is to highlight the uniqueness of the CPP and lessons that can be learned from the CPP by disaster risk management agencies worldwide. Among other things, we discuss unique features of the CPP like: the fact that the vast majority of its personnel consist of community volunteers, and its formal empowerment of women (to the extent of implementing a goal of 50% membership of women in the CPP). Another distinctive feature of the CPP is its implementation of a people-centered mode of risk communication wherein community residents are empowered to be risk communicators in their communities.

Comment:  Page 1, first paragraph: Are there any other natural hazards in Bangladesh besides cyclones and floods?

Response:   We have added the following sentence and citation.

In addition to the above, Bangladesh experiences other natural hazards such as drought, tornadoes, and riverbank erosion (Karim, 1995).

Karim, N. (1995). Disasters in Bangladesh. Natural Hazards, 11(3), 247-258.

Comment:  Page 2: Is this Program about helping people or is it about women taking leadership? Please write it more clearly and justify it. Will leadership by women solve the problem or not make it worse?

Response:  We have added the following text and citations.

Much literature is already available positively linking women's participation in disaster risk management to improved outcomes such as improved outreach, greater use of shelters, and reduced fatalities (e.g., UNISDR, 2009). There is ample literature documenting the greater vulnerability of lower-income women in parts of Bangladesh (e.g., Ikeda, 1995), and greater involvement and leadership of women in the CPP enables the organization to reach these vulnerable women more effectively (CPP, 2021b). And, so, the CPP helps communities by empowering women within them --it is a two-fold solution. It has been repeatedly seen in disasters that women and girls are most affected in disasters due to socio-economically and culturally disadvantaged positions in society (e.g., Thurston et al., 2021). So, to alleviate their plight, women’s engagement and leadership is imperative for a gender responsive approach to disaster response. 

CPP, Bangladesh Cyclone Preparedness Programme (2021b). WE-CPP: Women Empowerment in Cyclone Preparedness Programme, Ministry of Disaster Management and Relief, Dhaka, Bangladesh.

Ikeda, K. (1995). Gender differences in human loss and vulnerability in natural disasters: a case study from Bangladesh. Indian Journal of Gender Studies, 2(2), 171-193.

Thurston, A. M., Stöckl, H., & Ranganathan, M. (2021). Natural hazards, disasters and violence against women and girls: a global mixed-methods systematic review. BMJ global health, 6(4), e004377.

UNISDR (2009). Making Disaster Risk Reduction gender sensitive: policy and practical guidelines, United Nations International Strategy for Disaster Reduction, Geneva, Switzerland.

Comment:  Fig 1: Explain Y axes, what does left mean and what means right?

Response:  We have improved Figure 1 by adding labels to the left and right Y axes.

Comment:  The article lacks a thorough analysis of the phenomena and processes taking place. For example, isn't the fact that more women experience the effects of a cyclone because of the greater proportion of women in the population? Without such analyzes, the article is only a speculation by the authors.Is the preference for women to managerial positions not a symptom of discrimination against men? Should gender decide about performing managerial functions? Doubtful! Shouldn't you look for some other solutions here?

Response:  As described above (and in much of the text in the revised ms), in fact, it has been well established in the DRR literature that increased particiption by women is associated with improved disaster risk reduction. In the case of the CPP, increasing the numbers of women involved is not discriminating against men because traditionally, there were much greater numbers of men involved in the CPP volunteer corps than women. The present policy simply aims at reducing the disparity and, so, it is not a discrimination against men but a reduction in the shortage of women in the program (since roughly half of the community members are women, there is a shortage when much less than half of the volunteers are women --this is the gap that is being addressed). The numbers of female victims compared to men have been statistically shown in numerous studies (too numerous to cite in a short paper with a 2500 word limit), and these statistics are relative to the proportion of females to males in the surrounding population (e.g. use of odds-ratios). This is not speculation but a statistically demonstrated phenomenon --see, for example, the modified text in the revised ms below.

It is indeed important to understand that “disasters don’t discriminate, but people do… disasters reinforce, perpetuate and increase gender inequality, making bad situations worse for women” (UNISDR, 2009, 17). One historical review found that, in less developed countries, females had a higher risk of mortality during tropical cyclones (Doocy et al., 2013). One study (Nasreen, 2014) concluded that women in disaster prone areas often suffer more from sexual and reproductive health related problems during and after disaster. A study on violence against women in disasters (Nasreen, 2008) indicates that a large number (71.6%) of women (studied) were subject to more violence during disasters. Sexual harassment including forced sex, rape at home and in shelters were also reported by women and girls; and they do not want to take refuge in shelters during disasters due to lack of privacy and security. At the same time, women’s actual and potential contributions to disaster risk reduction is seldom recognised. Women’s role in strengthening a sustainability culture and their active contribution to household and community stability during disasters warrant due recognition.

Lastly, the primary points of the paper are summarized for clarity in the following added text.

A primary goal of the paper is to highlight the uniqueness of the CPP and lessons that can be learned from the CPP by disaster risk management agencies worldwide. Among other things, we discuss unique features of the CPP like: the fact that the vast majority of its personnel consist of community volunteers, and its formal empowerment of women (to the extent of implementing a goal of 50% membership of women in the CPP). Another distinctive feature of the CPP is its implementation of a people-centered mode of risk communication wherein community residents are empowered to be risk communicators in their communities.

Reviewer 2 Report

Dear Authors,

I just had the chance to read your work.

The format of this paper has extensive problems; for example, the reference section must be changed to meet thInternational Journal of Environmental Research and Public requirements.

Please refer to the template provided by the MDPI.

The novelty of this paper should be more elaborated. 

The discussion on the agenda must provide critical analysis of CPP's challenges and success factors as a disaster risk management agency compared with other agencies from a global perspective.

The contributions should be more highlighted. 

Also, the limitations of this work need more explanation.  

Kind Regards

Author Response

Response to Reviewer 2's Comments

The authors are grateful for the reviewer's comments and suggestions. As described below, we have endeavored to address all of them (while staying close to the journal's word limits for short articles).

Comment:  The format of this paper has extensive problems; for example, the reference section must be changed to meet thInternational Journal of Environmental Research and Public requirements.

 Please refer to the template provided by the MDPI.

Response:  We have reformatted the paper and put all references in a reference section at the end, in keeping with the style of the journal. The discussion was also expanded (see below), though the 2500 word limitation limited the quantity of additional text.

The novelty of this paper should be more elaborated. The contributions should be more highlighted. 

The following text has been added.

A primary goal of the paper is to highlight the uniqueness of the CPP and lessons that can be learned from the CPP by disaster risk management agencies worldwide. Among other things, we discuss unique features of the CPP like: the fact that the vast majority of its personnel consist of community volunteers, and its formal empowerment of women (to the extent of implementing a goal of 50% membership of women in the CPP). Another distinctive feature of the CPP is its implementation of a people-centered mode of risk communication wherein community residents are empowered to be risk communicators in their communities.

and, later in the paper:

Much literature is already available positively linking women's participation in disaster risk management to improved outcomes such as improved outreach, greater use of shelters, and reduced fatalities (e.g., UNISDR, 2009). There is ample literature documenting the greater vulnerability of lower-income women in parts of Bangladesh (e.g., Ikeda, 1995), and greater involvement and leadership of women in the CPP enables the organization to reach these vulnerable women more effectively (CPP, 2021b). And, so, the CPP helps communities by empowering women within them --it is a two-fold solution. It has been repeatedly seen in disasters that women and girls are most affected in disasters due to socio-economically and culturally disadvantaged positions in society (e.g., Thurston et al., 2021). So, to alleviate their plight, women’s engagement and leadership is imperative for a gender responsive approach to disaster response. 

CPP, Bangladesh Cyclone Preparedness Programme (2021b). WE-CPP: Women Empowerment in Cyclone Preparedness Programme, Ministry of Disaster Management and Relief, Dhaka, Bangladesh.

Ikeda, K. (1995). Gender differences in human loss and vulnerability in natural disasters: a case study from Bangladesh. Indian Journal of Gender Studies, 2(2), 171-193.

Thurston, A. M., Stöckl, H., & Ranganathan, M. (2021). Natural hazards, disasters and violence against women and girls: a global mixed-methods systematic review. BMJ global health, 6(4), e004377.

UNISDR (2009). Making Disaster Risk Reduction gender sensitive: policy and practical guidelines, United Nations International Strategy for Disaster Reduction, Geneva, Switzerland.

Comment:  The discussion on the agenda must provide critical analysis of CPP's challenges and success factors as a disaster risk management agency compared with other agencies from a global perspective.

Response:  The following modified text is found in the revised ms (see especially text at the end).

It would be good to compare the CPP to similar agencies in other cyclone-prone countries in this region. In the case of India, disaster management falls under the purview of the national agency, the National Disaster Management Authority (NDMA), which extends to the state level (State Disaster Management Authority, or SDMA) and districts. This agency disseminates early warning at the lower levels and carries out evacuation in coordination with Civil Defense.  Similarly, the Philippines implements disaster management in hierarchical fashion, from the National Disaster Risk Reduction and Management Council (NDRRMC) to Provincial Disaster Risk Reduction and Management Council (PDRRMC) to Local Disaster Risk Reduction and Management Council (LDRRMC), and shares responsibilities for disseminating early warnings with the Office of Civil Defense, while evacuation is mainly carried out by the LDRRMC (in coordination with local government units and community-based organizations). Similarly, Pakistan's disaster management program has a hierarchical structure, from the national NDMA to PDMA to DDMA, and early warnings are course through this chain of command. These agencies coordinate with Civil Defense, which works with local district governments in evacuation. 

The CPP is distinctive in several respects. First, it is a single-purpose agency with a special focus on tropical cyclones. This may lend to a simpler organizational arrangement, in which all functions related to cyclone preparedness and response are centralized within the CPP. While we do not have enough data to analyze the organizational strengths and limitations of this more streamlined organizational design vis-a-vis other national agencies, suffice it to say that this arrangement tends to allow easier replicability from district to district.

The other distinctive feature of the CPP is its formal commitment to community-based disaster management (CBDM). Note that other nation's agencies also have a commitment to CBDM, but in less formal fashion. For example, in India, community volunteers are organized on an ad hoc level --in some cases, through Civil Defense and, in others, through different local and regional civic organizations.  A similar arrangement is found in the Philippines, where various NGOs and CBOs coordinate with the Office of Civil Defense and LDRRMCs in a manner that differs from locality to locality. The CPP is unique in the extent to which community-based disaster management is formalized --most of all evident in the fact that most of the workforce of CPP is composed of formally deputized CPP volunteers. The formal deployment of community residents translates into a formal commitment to gender equity as well --the CPP is also unique among similar natonal agencies in its formal goal of 50% participation by women in the CPP volunteer corps , as discussed below. It is quite possible that the strength of these commitments to community participation and gender equity are fostered by the clarity of CPP's mandate and organization...

The CPP, more so than other international agencies, has succeeded in creating a largely volunteer-led corps for early warning and emergency preparedness and in institutionalizing parity of participation by women. These also constitute ongoing challenges in maintaining the CPP. For example, there is often a discussion about whether the volunteer corps should be given, if not formal salaries, stipends, but this brings up the issue of whether participation would then be driven more by financial incentives than the motivation to help.

Comment:  Also, the limitations of this work need more explanation.  

Response:  We have added the following text to the conclusion section (albeit we were only able to add so much text because of the word limits for a short article).

The foremost limitation in assessing organizational performance in disaster risk preparedness is the inability to statistically correlate improved outcomes with organizational efforts. There are no controlled experimental conditions possible in disaster risk management. As such, monitoring progress (as seen in Figure 1) is valuable, but formally correlating this with different organizational features and actions is not something that can be strongly established statistically.

Round 2

Reviewer 1 Report

The authors significantly improved the text.

Reviewer 2 Report

Accept in present form